# The structural basis of tRNA recognition by arginyl-tRNA-protein transferase

Thilini Abeywansha[1,5], Wei Huang [2,5], Xuan Ye [1,3], Allison Nawrocki[1], Xin Lan[1], Eckhard Jankowsky [1,3,4], Derek J. Taylor [1,2,4] ✉ & Yi Zhang [1,4] ✉

Arginyl-tRNA-protein transferase 1 (ATE1) is a master regulator of protein homeostasis, stress response, cytoskeleton maintenance, and cell migration. The diverse functions of ATE1 arise from its unique enzymatic activity to covalently attach an arginine onto its protein substrates in a tRNA-dependent manner. However, how ATE1 (and other aminoacyl-tRNA transferases) hijacks tRNA from the highly efficient ribosomal protein synthesis pathways and catalyzes the arginylation reaction remains a mystery. Here, we describe the three-dimensional structures of *Saccharomyces cerevisiae* ATE1 with and without its tRNA cofactor. Importantly, the putative substrate binding domain of ATE1 adopts a previously uncharacterized fold that contains an atypical zinc-binding site critical for ATE1 stability and function. The unique recognition of tRNA[Arg] by ATE1 is coordinated through interactions with the major groove of the acceptor arm of tRNA. Binding of tRNA induces conformational changes in ATE1 that helps explain the mechanism of substrate arginylation.

Arginyl-tRNA-protein transferase 1 (ATE1) is a 58 kDa aminoacyl transferase that is conserved among nearly all eukaryotes. With the aid of a transfer RNA (tRNA) cofactor, ATE1 covalently adds arginine (arginylation) to its protein substrates to promote their degradation[1–3]. Homozygous ATE1 knockout in mice induces embryonic death with severe defects in cardiovascular development and angiogenesis[1], while conditional ATE1 knockout leads to reproductive defects, growth/ behavioral retardation, enhanced clot retraction, and brain abnormalities in mammals/vertebrates[4–7]. Although not essential for yeast viability, ATE1 is required for physiological responses to stress conditions to induce growth arrest and prevent stress-induced DNA mutagenesis[8]. The broad roles of ATE1, including cell growth and migration, stress responses, and nerve regeneration[9–12], are executed mainly through arginylation of diverse substrates that alters protein turnover[3,11,13]. Specifically, arginylation was identified as a destabilizing modification on the N-terminal residue of several proteins, which decreased protein half-life to a few minutes by evoking poly-ubiquitination and degradation through the ubiquitin-proteasome system[11,13]. Later studies, including work from our group, determined

that arginylated proteins are also degraded through SQSTM1/p62-dependent selective autophagy[14,15]. Interestingly, recent studies also revealed non-degrative roles of N-terminal arginylation and identified non-canonical midchain arginylation sites in many proteins, implicating even broader roles of arginylation in cells[16–19].

Chemically, N-terminal arginylation involves the formation of a peptide bond between the carboxyl group of arginine and the amino group of aspartate/glutamate (Fig. 1a). The reaction requires arginyl-tRNA[Arg] as the arginine donor, and ATE1 catalyzes the specific transfer of arginine to the first residue (Asp or Glu) of substrates. Once the transfer is complete, tRNA[Arg] is released and recharged by arginyl-tRNA synthetase (ArgRS). The arginylation reaction is ATP-independent and can be reconstituted in vitro[20]. This unusual catalytic mechanism places ATE1-catalyzed-protein-arginylation in one of the rare categories where a ribosomal aminoacyl-tRNA is used outside of protein synthesis.

Despite its biological significance and unique enzymatic properties, little is known about the catalytic mechanism and regulation of ATE1. Recent studies demonstrate that mouse ATE1 selectively binds to

[1]Department of Biochemistry, Case Western Reserve University, Cleveland, OH 44106, USA. [2]Department of Pharmacology, Case Western Reserve University, Cleveland, OH 44106, USA. [3]Center for RNA Science and Therapeutics, School of Medicine, Case Western Reserve University, Cleveland, OH 44106, USA. [4]Case Comprehensive Cancer Center, School of Medicine, Case Western Reserve University, Cleveland, OH 44106, USA. [5]These authors contributed equally: Thilini Abeywansha, Wei Huang. ✉e-mail: djt36@case.edu; yi.zhang26@case.edu

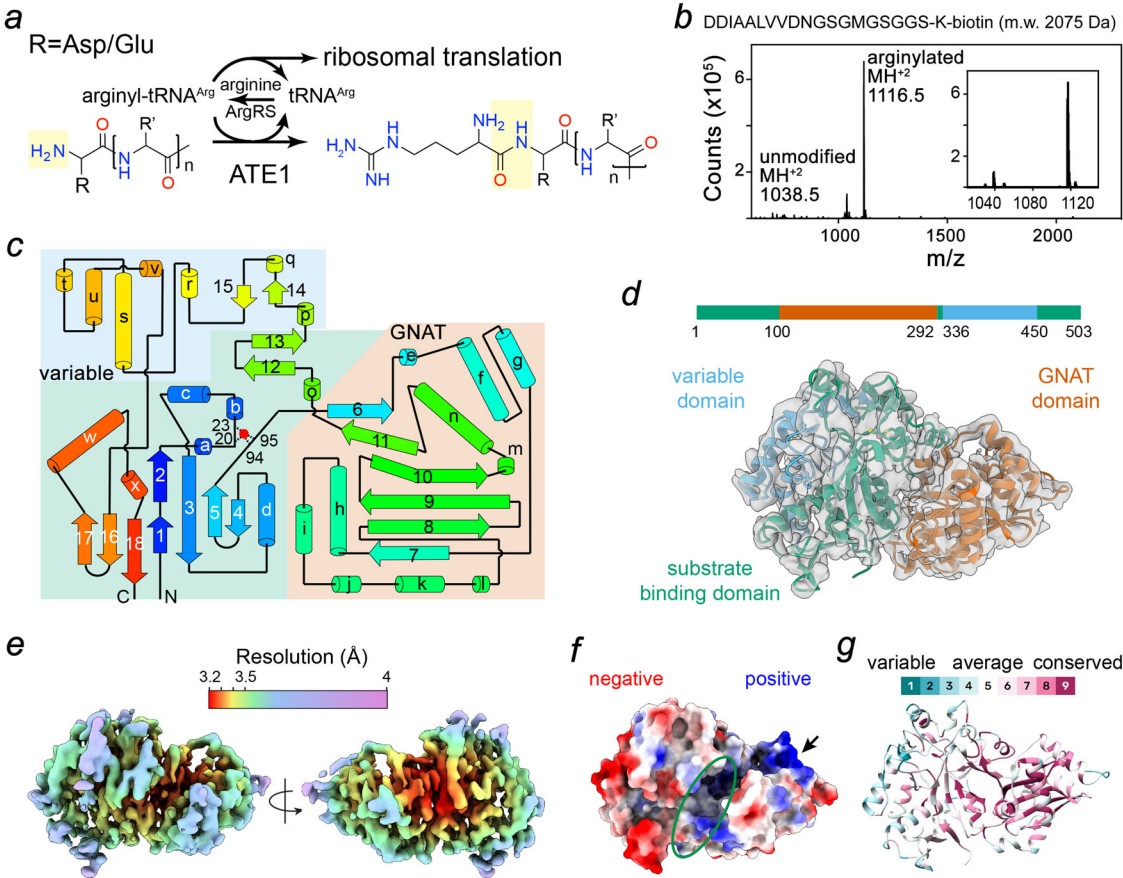

**Fig. 1 | Protein arginylation reaction and overall structure of *S. cerevisiae* ATE1.** **a** Arginyl-tRNA[ARG]-dependent N-terminal arginylation of polypeptides containing aspartates or glutamates. **b** In vitro arginylation of the indicated substrate peptide using recombinantly purified *sc*ATE1 monitored by liquid chromatography–mass spectrometry (LC–MS) analysis. **c** 2D topological diagram of *sc*ATE1 with α-helices indicated by cylinders and β-strands by arrows. **d** The cryo-EM structure of *sc*ATE1 (3.2 Å). Ribbon representation of *sc*ATE1 with the color-coded domain organization on a linear sequence: GNAT domain (vermillion), substrate binding domain (bluish green) and variable domain (sky blue). **e** Local resolution of *sc*ATE1 displayed on the cryoEM map. The color code is shown in the color bar. **f** Surface representation of *sc*ATE1 colored based on electrostatic potential. Blue and red for positive and negative charges, respectively. The putative substrate binding site is highlighted in a green oval. **g** The ribbon diagram of *sc*ATE1 colored based on the sequence conservation using ConSurf server. For visualization of conservation, the most variable positions (grade 1) were colored turquoise, and the most conserved positions (grade 9) were colored maroon per Consurf coloring scheme.

arginyl-tRNA[21]; however, the molecular mechanism by which ATE1 interacts with, and selects for, arginyl-tRNA[Arg] remains elusive. Here, we report two structures of ATE1 from *Saccharomyces cerevisiae* (*sc*ATE1) in apo form and in complex with tRNA[Arg]. The structure of *sc*ATE1 displays an overall new fold with a compact conformation consisting of three domains, including a highly conserved GCN5-related N-acetyltransferases (GNAT) fold, a putative polypeptide substrate binding domain, and a unique variable domain. Furthermore, we identified an atypical zinc finger motif in the putative substrate binding domain that enhances ATE1 stability and function. In the assembled complex, ATE1 interacts with tRNA[Arg] via an α-helix that forms molecular contacts with the major groove of the tRNA[Arg] acceptor arm, all of which results in structural rearrangements of the ATE1 protein. These rearrangements most likely facilitate the transfer of arginine from the tRNA cofactor to the protein substrate.

## Results

### Overall Structure of *Saccharomyces cerevisiae* ATE1

To elucidate the molecular basis of the arginylation reaction, we expressed and purified *sc*ATE1 recombinantly for structural studies. Since mouse ATE1 undergoes self-arginylation[20], we first used liquid chromatography-mass spectrometry (LC-MS) to assess whether *sc*ATE1 is similarly modified. Analyses revealed a single, major peak (57,956 Da) for *sc*ATE1 protein nearly identical to its theoretical mass

(57,954 Da), thereby indicating no self-arginylation or other covalent modification occurred. The ability of mouse ATE1 to use *E. coli* or mouse tRNA[Arg] for catalysis[21] suggests that ATE1 does not select species-specific tRNA[Arg]. To confirm this hypothesis, we performed a reconstituted arginylation assay using purified *sc*ATE1, in vitro transcribed human tRNA[Arg], and human ArgRS (arginyl-tRNA synthetase). LC-MS analysis of the reaction mixture identified an arginylated product with the expected mass increase of 156 Da (mono-arginylation), thereby confirming that the recombinantly produced *sc*ATE1 is active even with human tRNA[Arg] as a cofactor (Fig. 1b).

We first determined the structure of full-length *sc*ATE1 using single-particle cryo-electron microscopy (cryoEM) to an average resolution of 3.2 Å. *Sc*ATE1 folds into a compact and globular conformation consisting of 18 well-defined β-strands along with 15 long and 9 short α-helices (Fig. 1c–e and Supplementary Table 1). Structural comparison using the DALI program[22] against known structures in the protein database (PDB) revealed low overall similarity with the best hit showing a Z-score of 8.8 (PDB: 4v36) covering ~60% of *sc*ATE1 sequence. The aligned region displays the topology pattern of a typical GNAT fold consisting of a mixed α/β/α structure[23]. DALI analysis for the rest of the structure returned no hits with a Z-score (typical range 2-60) above 4, indicating a new fold that is not represented in the PDB. We noted that two crystal structures of apo ATE1, which are consistent with our cryo-EM structure, were released while this study was under review (see discussion).

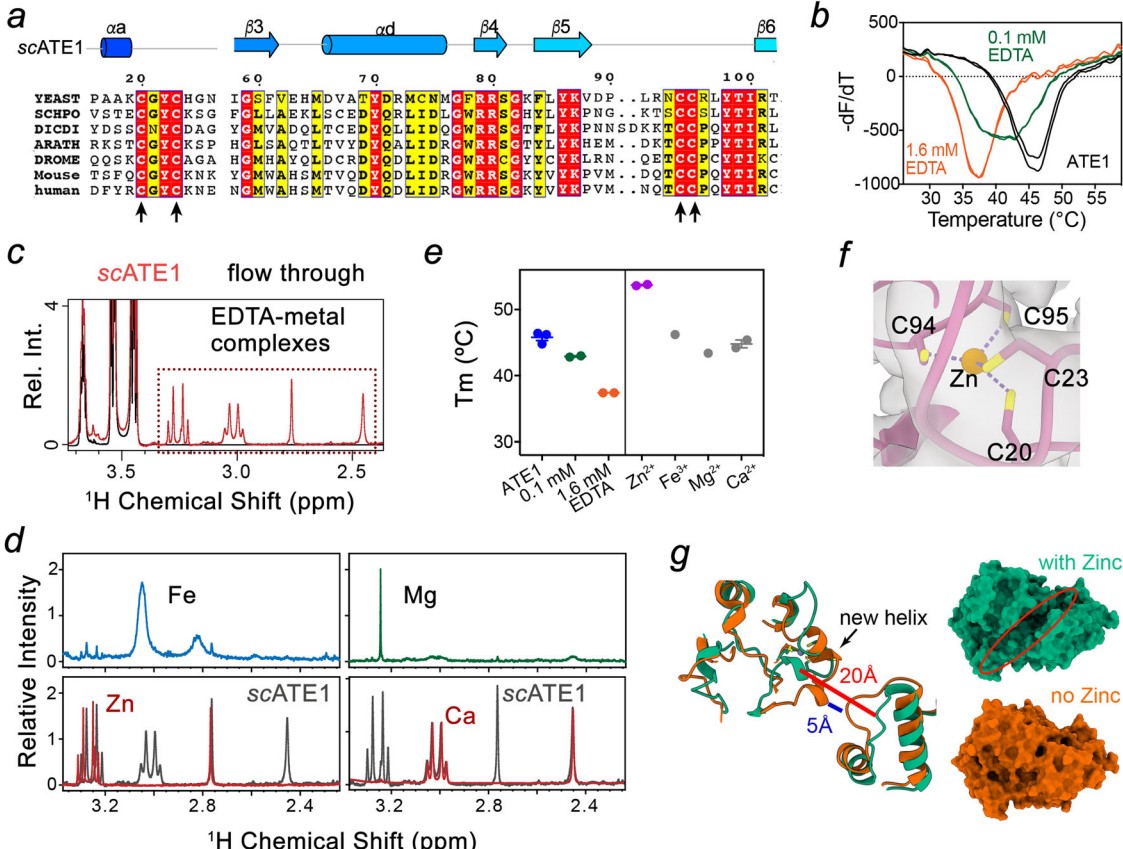

**Fig. 2 | Identification of an atypical zinc-finger in *sc*ATE1. a** Multiple sequence alignment for the N-terminal region of curated ATE1 sequences from model organisms using Clustal Omega and displayed using ESPript3. Arrows and cylinders indicate β-strands and α-helices, respectively. Red and yellow backgrounds indicate identical and similar residues, respectively. **b** ATE1 stability upon addition of EDTA monitored by DSF. The first negative derivative of fluorescence intensity is plotted against temperature. The melting temperature ($T_m$) is defined as the inflection point of the melting curve, corresponding to the peak. **c** Superimposed ¹H NMR spectra of EDTA-dissolved powder prepared using lyophilized ATE1 protein sample (red) and flow through (black) from the centrifugal filter unit. **d** Superimposed ¹H NMR spectra of EDTA-metal complexes from ATE1 protein sample and EDTA-cation standards. **e** $T_m$ values derived from DSF for the ATE1 protein in the presence of EDTA, $Zn^{2+}$, $Fe^{3+}$, $Mg^{2+}$, and $Ca^{2+}$ ions. Error bars represent SEM determined from triplicate or duplicate independent experiments ($n = 3$ or 2). **f** A zoom-in view of the putative ion binding site coordinates four cysteines, 20, 23, 94 and 95.
**g** Superposition of two snapshots (t = 300 ns) from both trajectories: with (green) and without (orange) zinc. Distance between helix αa and the loop between helix αh and αi is highlighted for each structure: 20 Å in the presence of zinc (red line) and 5 Å in the absence of zinc (blue line). Surface representation of each structure is shown on the left. The substrate binding cleft is highlighted in a red oval, which closes in the absence of zinc. Source data are provided as a Source Data file.

Within this new fold, the N- and C- termini of *sc*ATE1 coalesce (Fig. 1d), forming a mixed β-sheet (β1-β18-β16-β17 and β2-β3-β5-β4) facing the GNAT domain. Analysis of ATE1 sequences from model organisms predicts that only the mixed β-sheet and the GNAT domain are shared across species (Supplementary Fig. S1). Notably, the electrostatic surface potential of the *sc*ATE1 defines a large positively charged surface area on the GNAT domain that could interact with negatively charged nucleic acid (Fig. 1f). Consistently, mapping amino acid sequence conservation across species onto the structure of *sc*ATE1 revealed that the GNAT domain and the cleft between the GNAT and substrate binding domains are highly conserved (Fig. 1f, g). As such, we refer to the conserved and variable domains as the putative substrate binding and variable domains, respectively (Fig. 1d).

### ATE1 N-terminal domain contains an atypical zinc-finger

Inspection of the cryoEM map with the modeled structure revealed density within the substrate binding domain that cannot be solely accounted for by protein sidechains. This density localizes to a highly conserved cystine cluster, comprised of C20, C23, C94 and C95, and is indicative of metal coordination (Fig. 2a). To determine whether ATE1 is a metalloprotein, we examined the thermal stability of *sc*ATE1 by

differential scanning fluorimetry (DSF) in the presence and absence of the chelating agent EDTA. The melting temperature of *sc*ATE1 (46 °C) decreased with increasing concentrations of EDTA, suggesting that divalent metal ions are required to stabilize ATE1 (Fig. 2b). Since EDTA can extract ions from *sc*ATE1, we sought to determine the identity of metal ions by their characteristic proton chemical shifts within the metal-EDTA complex using nuclear magnetic resonance (NMR) spectroscopy[24]. Compared to the flow-through sample collected after ultrafiltration, *sc*ATE1 sample clearly showed characteristic signals of EDTA-zinc and EDTA-calcium but not EDTA-magnesium nor EDTA-iron complexes (Fig. 2c, d). Consistently, supplementing additional zinc ions further stabilized *sc*ATE1, as evidenced by an 8 °C shift in the thermal melting curve (Fig. 2e). This finding is indicative of a weak zinc binding of ATE1 with partial metal cofactor loss during purification. Of note, calcium was also present in our EDTA control and is a known contaminant of plasticware and glassware[25]. Additionally, calcium ions are almost exclusively coordinated by oxygen ligands from Asp, Glu, Asn, Gln, Ser, and Thr residues in proteins[26]. Therefore, the observed calcium ion in ATE1 was unlikely to be the ligand for the cystine cluster. Together, our results strongly suggest that a zinc ion is the metal ligand of the cystine cluster and is important for protein stability (Fig. 2f).

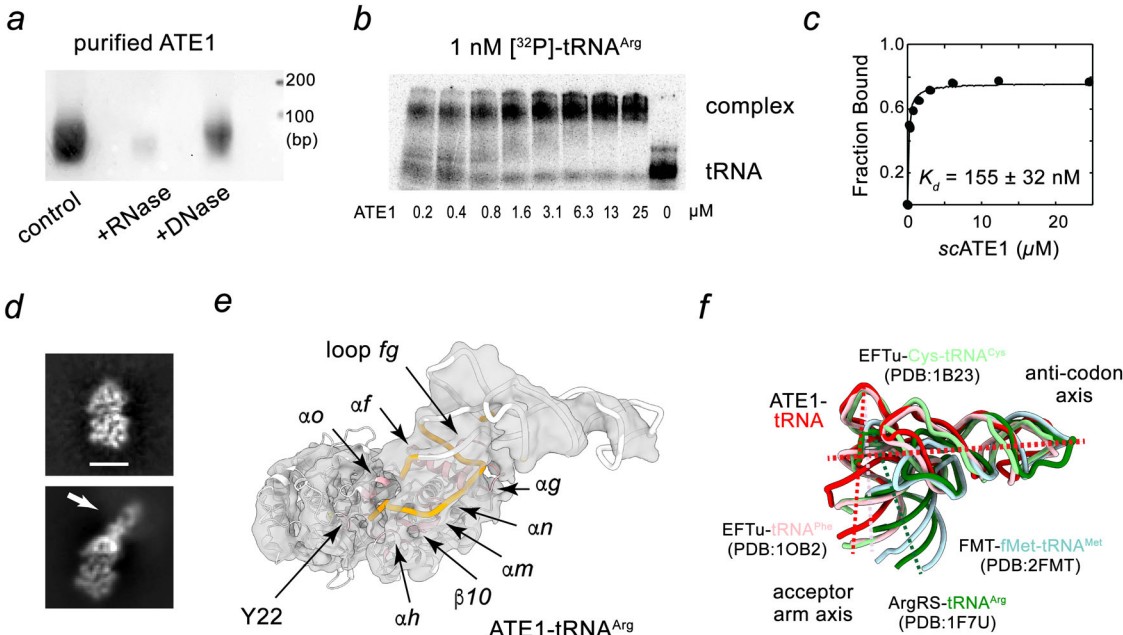

**Fig. 3 | Interactions between *sc*ATE1 and tRNA. a** Agarose gel electrophoresis of purified *sc*ATE1 protein sample with or without DNase or RNase treatment. At least three independent experiments were performed with similar results. **b** EMSA visualization and **c** quantification of ATE1 binding to in vitro transcribed tRNA[Arg] labeled with [32]P. **d** Representative 2D class averages of ATE1 particles (top) and ATE1 with density features indicative of bound nucleic acid (bottom; highlighted with arrow). Scale bar is 50 Å. **e** CryoEM structure of *sc*ATE1-tRNA complex. Protein interface is colored in pink and RNA residues contacting protein are highlighted in orange. Annotation of individual secondary structure elements are labeled. **f** Superposition of different tRNA structures aligned to their anti-codon axis. These tRNA complexes include EF-Tu •tRNA[Phe] (PDB entry: 1OB2), EF-Tu •tRNA[Cys] (PDB entry: 1B23), methionyl-tRNA[fMet] transformylase (PDB entry: 2FMT) and arginyl-tRNA synthetase •tRNA[Arg] (PDB entry: 1F7U). Anti-codon axis is indicated by a red, horizontal dashed line, and acceptor arm axes are highlighted by dashed lines that are colour-coded based on labels. Source data are provided as a Source Data file.

Because ATE1 was not previously known to be a metalloprotein, we performed further analysis regarding the zinc-binding motifs containing the 'CX$_2$CX$_n$CC' sequence. To the best of our knowledge, no such sequence and structure determinants (loop$_{CXXC}$-$\beta$-$\alpha$-$\beta$-loop$_{CC}$) has been reported for zinc fingers (ZnFs) or other metal-binding proteins. ZnFs have versatile functions and mediate diverse interactions with DNA, RNA, protein, and/or lipid substrates[27–31]. To delineate the function of the metal binding in ATE1, we performed molecular dynamics (MD) simulations of *sc*ATE1 in the presence and the absence of a zinc atom in the metal center. The protein fold in both simulations was maintained, with deviations of around 3 Å root-mean-square-deviation (RMSD) as compared to the starting cryoEM structure. However, in the absence of zinc, the simulation resulted in protein changes with slightly higher RMSD values as compared to simulations performed in the presence of zinc (Supplementary Fig. S3a, b). In these simulations, the radius of gyration (Rg) of *sc*ATE1 without zinc increases ~0.5 Å in the first 70 ns before shifting back to a similar size as that conducted with zinc (Supplementary Fig. S3c). Examining the Rg for the four-cysteine cluster reveals that zinc holds these four cysteines together (Supplementary Fig. S3d). A snapshot from the last 100 ns trajectory without zinc reveals that local geometry disruption of the ZnF leads to the formation of a new short helix, altering the relative bending angle between β1 and β2 strands of ATE1 (Fig. 2g). This conformational change leads to the disruption of the putative substrate binding cleft (Fig. 2g). Interestingly, the variable domain demonstrates the most increase in per residue root-mean-square-fluctuations (RMSF) when zinc is omitted from the simulation (Supplementary Fig. S3e). The increased flexibility observed in the variable domain that correlates with the absence of zinc in the simulations could explain the decrease in thermal stability of ATE1 when zinc was omitted from the experiments. Collectively, our results demonstrate a critical role of the ATE1-ZnF in maintaining ATE1 stability and suggest that metal binding is a mediator of polypeptidyl-substrate access and binding to ATE1.

## The *sc*ATE1•tRNA[Arg] complex structure

ATE1 is highly specific and exclusively transfers arginine using arginyl-tRNA[Arg] as the donor cofactor[21]. Incubation of mouse ATE1 with extracted total mouse liver RNAs enriches for tRNA[Arg] species, while other cellular tRNAs are under-presented with only a few exceptions[21]. Therefore, we sought to determine whether any endogenous tRNA might co-purify with the recombinant *sc*ATE1 protein. The purified protein correlated with an elevated A$_{260}$/A$_{280}$ ratio even after size exclusion chromatography, indicating the existence of nucleic acid components bound to *sc*ATE1. These nucleic acids were sensitive to RNase but not DNase treatment, consistent with the role of ATE1 in recognizing RNA (Fig. 3a). To further characterize the putative ATE1-tRNA interaction, we measured gel shifts with radiolabeled tRNA[Arg], which maintains its ability to mediate the arginylation reaction (Fig. 1b). These data revealed that the recombinant *sc*ATE1 bound tRNA[Arg] with an apparent dissociation constant (155 nM), indicating an ability of ATE1 to bind tRNA[Arg] (Fig. 3b, c).

We thus rationalized that we may obtain ATE1•tRNA[Arg] complex suitable for structural interrogation using cryoEM by co-overexpressing tRNA[Arg] with *sc*ATE1. To this end, we used *E. coli* BL21 (DE3)-RIL cells for protein production, which encodes a rare codon plasmid containing an extra copy of *argU* and overexpresses tRNA[Arg]. The purified sample was heterogenous and contained both free ATE1 protein as well as complexes with additional density that could be attributed to tRNA-binding to ATE1. Specifically, upon sorting, a population of particles emerged with density features of major and minor grooves resembling duplex nucleic acid in 2D class averages (Fig. 3d and Supplementary Fig. S2b). An additional round of 3D classification was implemented to identify a subclass of particles with strong map features that could be attributed to tRNA. Due to the presence of preferred orientations, the gold-standard FSC curve of 3.7 Å is an over-estimated resolution for the cryoEM map (Supplementary Fig. S2c–f). However, the major and minor grooves of the

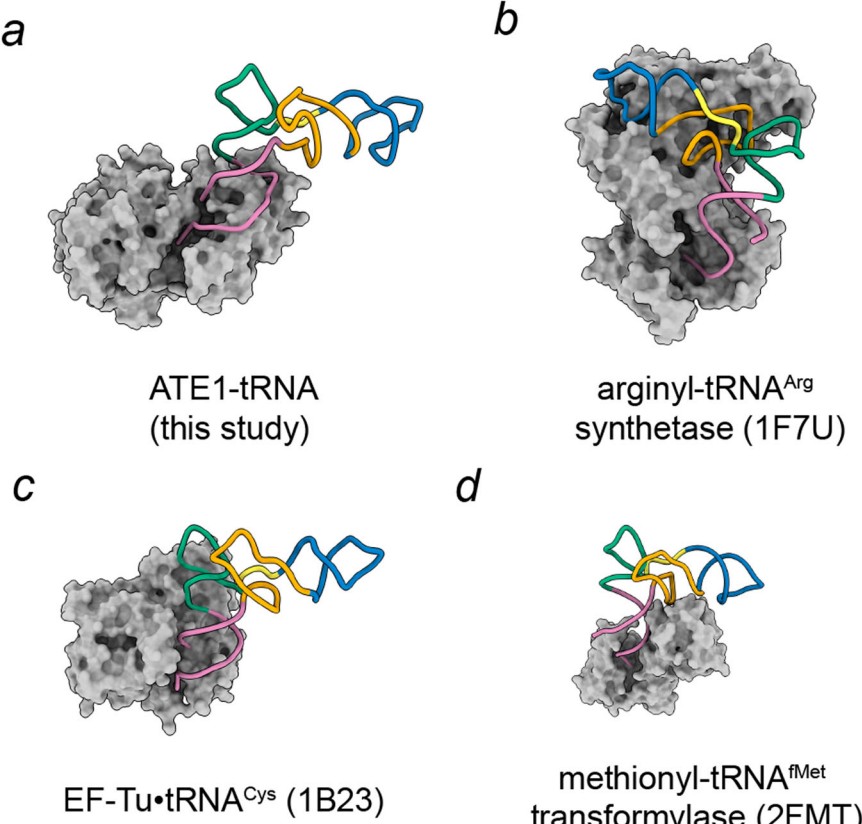

**Fig. 4 | Structural comparison of tRNA bound complexes. a** *Sc*ATE1• tRNA$^{Arg}$ complex in this study. **b** Arginyl-tRNA synthetase•tRNA$^{Arg}$ complex (PDB entry: 1F7U). **c** EF-Tu •tRNA$^{Cys}$ complex (PDB entry: 1B23). **d** Methionyl-tRNA$^{fMet}$ transformaylase complex (PDB entry: 2FMT). The tRNA molecules are colored based on specific regions: acceptor arm (reddish purple), T-arm (bluish green), variable loop (yellow), D-arm (orange), anticodon arm (blue).

bound tRNA and protein α-helix features are well resolved. Although the resolution of the current map does not allow for a detailed analysis of the residue-specific interactions, the general feature of ATE1-tRNA association can still be confidently defined. In ATE1, helix αf inserts into the major groove of the tRNA acceptor stem. Based on the geometry and canonical tRNA structures, it is evident that the tRNA acceptor arm is contacting ATE1 (Fig. 3e).

To further extrapolate interactions between *sc*ATE1 and tRNA from the map, we performed all-atom molecular dynamic flexible fitting (MDFF) with explicit solvent model (see methods) to obtain structure models for the assembled complex. In this *sc*ATE1•tRNA$^{Arg}$ structure, it is evident that the GNAT domain is the major interaction site with tRNA (Fig. 3e). In particular, the interacting region primarily involves αf, the first turn of helix αg and their connecting loop (a.a. 109-146). Additional tRNA interacting sites on ATE1 include the C-terminal end of helix αh (a.a. 170-178), β10 and the N-terminus of helix αn, and helix αm (a.a. 257-271). Bound tRNA is also proximal to helix αo (a.a. 299-304) and Y22, both of which contribute to the substrate binding domain of *sc*ATE1. To compare the tRNA conformation observed in *sc*ATE1•tRNA$^{Arg}$ with other tRNA-bound complexes, we superimposed tRNAs along their anti-codon axis, revealing a spectrum of angles that the acceptor arm adopts relative to the anti-codon arm (Fig. 3f). The structural comparison highlights the flexibility of tRNA as it adopts multiple, but distinct, conformations to accommodate binding of diverse protein partners (Fig. 4). Proteins such as tRNA synthetases contact both the acceptor and anti-codon arms to confine tRNA in a conformational state with a smaller angle between the acceptor arm axis and anti-codon axis. Other proteins, such as ATE1 and elongation factors, contact only one axis of the "L"-shape of tRNA, thereby allowing it to adopt a wider angle between the two axes. Structurally,

this wider angle fully exposes the 3′-strand of the acceptor arm to be detected by ATE1 and anchoring the 3′-RCCA towards the enzyme active site. It is likely that this angle between two arms of the L-shape could be encoded by sequences within the variable loop of different tRNAs. Therefore, our observation hints at the possibility that ATE1 recognition of tRNA involves three-dimensional shape selection.

### Molecular mechanism of the ATE1 association with the acceptor arm of tRNA$^{Arg}$

Our *sc*ATE1•tRNA$^{Arg}$ structure provides a vivid illustration to explain how an *E. coli* tRNA$^{Arg}$-derived miniRNA hairpin containing only the acceptor arm is sufficient to mediate arginylation[21]. To further characterize the association between ATE1 and the tRNA acceptor arm, we synthesized the same miniRNA hairpin and examined its interaction with *sc*ATE1 (Fig. 5a). Indeed, the addition of *sc*ATE1 shifts the free miniRNA to the complex form with a sub-μM affinity. In contrast, a DNA oligo with the corresponding sequence only weakly associates with *sc*ATE1, suggesting that the A-form geometry and/or the 2′-hydroxyl group of tRNA is important for the specific recognition by ATE1. These results were further corroborated by quantitative measurement of the binding affinities using microscale thermophoresis (MST) assays (Fig. 5b and Supplementary Fig. S4a). ATE1 bound to fluorescently labeled DNA oligo with a $K_d$ ~ 35 μM, whereas binding to the tRNA acceptor arm was ten-fold stronger ($K_i$ ~ 3.6 μM). Altogether, our biochemical data further confirm that the acceptor stem of tRNA is sufficient to maintain interactions with *sc*ATE1.

Our structural analysis indicates that every nucleotide in the 3′-strand of the tRNA acceptor arm is in close contact with *sc*ATE1, highlighting the role of precise positioning of the tRNA 3′-end into the enzyme active site. Interestingly, A77 of tRNA is in close contact with

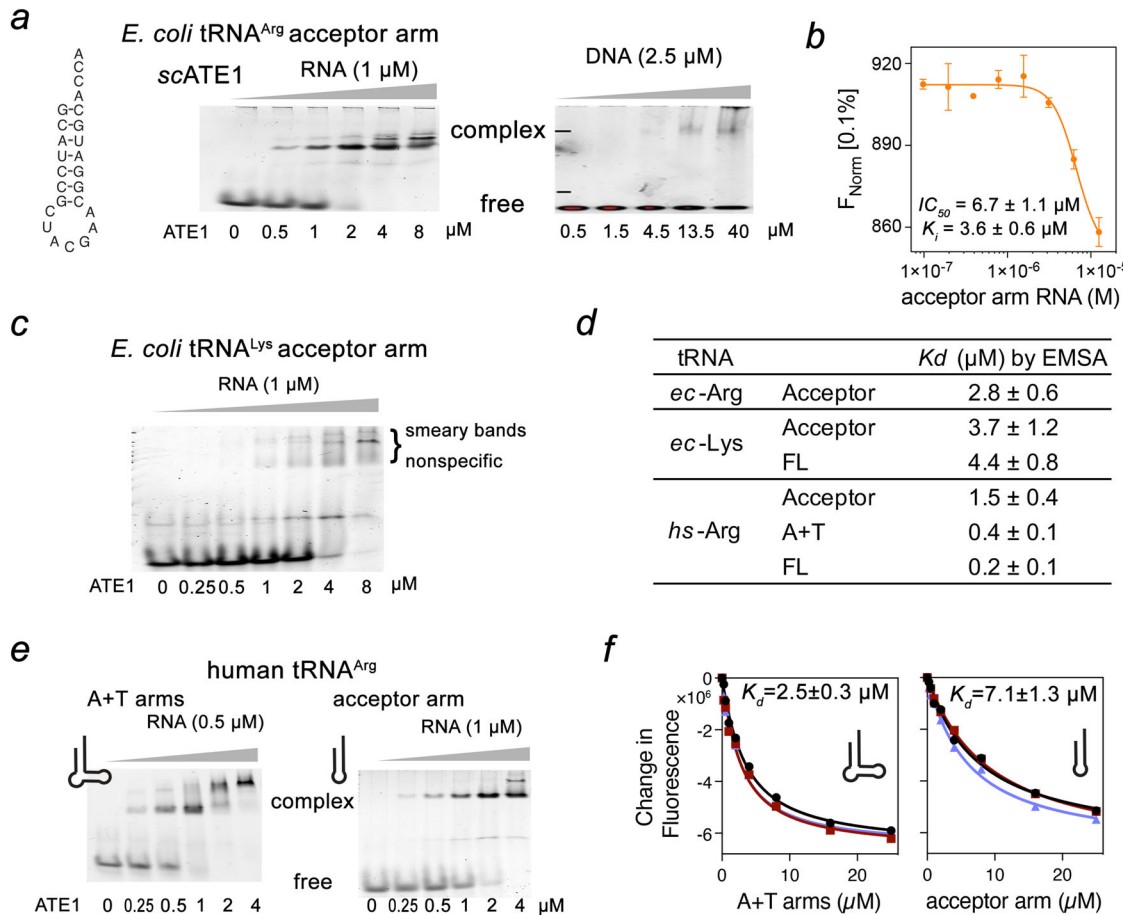

**Fig. 5 | Acceptor arm recognition and the proposed mechanism of ATE1-mediated arginylation. a** *Sc*ATE1 recognizes the acceptor arm of tRNA. Representative EMSA assay of ATE1 binding to synthesized RNA (middle) and DNA (right) with sequence shown on the left. ATE1 concentrations are shown below the gel. Three independent experiments were performed with similar results. **b** Binding curves used to determine the $K_i$ value of ATE1-RNA interaction in a competition assay by MST. Error bar represents s.d. in duplicate measurements. **c** Representative EMSA assay of ATE1 binding to synthesized tRNA$^{Lys}$ acceptor arm. Three independent experiments were performed with similar results. **d** $K_d$ values of

ATE1-RNA interaction in EMSA assays. Error represents SEM of at least two separate experiments. **e** Representative EMSA assay of ATE1 binding to synthesized human tRNA$^{Arg}$ A + T arms and acceptor (A) arm alone. Three independent experiments were performed with similar results. **f** Binding curves used to determine the $K_d$ values by fluorescence spectroscopy. Data points and fitted curves for three independent experiments are shown and colored black, red, and blue. Data are presented as mean values +/− SD in triplicate measurements. Source data are provided as a Source Data file.

Y22 of *sc*ATE1, which is located in the loop between the zinc coordinating C20 and C23. This finding suggests that the ZnF of *sc*ATE1 plays a direct role in facilitating tRNA recognition and binding. We, therefore, performed EMSA experiments with miniRNA and EDTA to evaluate the contribution of the ZnF in tRNA binding. Disruption of the ZnF led to a decrease in RNA binding compared to the untreated ATE1 (Supplementary Fig. S4b), highlighting the importance of ATE1-ZnF in maintaining a channel for both substrate and arginyl-tRNA to bind.

Does ATE1 select tRNA$^{Arg}$ from other tRNAs? To answer this question, we synthesized the acceptor arm of *E. coli* tRNA$^{Lys}$ and analyzed its association with *sc*ATE1. Compared with the *E. coli* tRNA$^{Arg}$ acceptor arm, tRNA$^{Lys}$ acceptor arm bound ATE1 only slightly weaker with a $K_d$ value of 4 μM (Fig. 5c, d), as quantified by EMSA experiments. However, the complex formed by tRNA$^{Lys}$ acceptor arm and ATE1 exhibited smeary bands, possibly indicating non-specific binding (Fig. 5c). Interestingly, *sc*ATE1 bound to the human tRNA$^{Arg}$ acceptor arm 2-fold stronger (1.5 μM) than its *E. coli* counterpart, suggesting that ATE1 has preferred tRNA$^{Arg}$ variants despite being able to broadly use tRNA$^{Arg}$ from different species for enzyme activity. Moreover, as seen in the *sc*ATE1•tRNA$^{Arg}$ complex structure (Fig. 4a), the T-arm of tRNA$^{Arg}$ is also proximal to ATE1, which likely provides additional interactions to stabilize binding. Indeed, synthesized RNA comprising both the A-

and T-arms (denoted as A + T arms) of tRNA$^{Arg}$ enhanced ATE1 binding affinity by 4-fold, which is comparable to the binding affinity of full-length tRNA (Fig. 5d–f). In contrast, full-length tRNA$^{Lys}$ bound to ATE1 with an affinity similar to that of the acceptor arm alone (Fig. 5d and Supplementary Fig. S4c), further substantiating that the A + T arms of tRNA confer the selectivity of ATE1 toward tRNA$^{Arg}$. Collectively, our biochemical and structural characterization of *sc*ATE1 interactions with tRNA and acceptor arms provide a molecular depiction of the tRNA recognition mechanism.

### Recognition of arginyl-tRNA$^{Arg}$ and catalytic cycle of protein arginylation reaction

The majority of cellular tRNAs are charged, at least under normal nutrient conditions[32,33]. Therefore, the chemical nature of the tRNA-bound amino acid moiety should also be critical for specific recognition of arginyl-tRNA$^{Arg}$ by ATE1. This notion is supported by data showing that ATE1 cannot transfer a lysine from a mischarged lysyl-tRNA$^{Arg}$ or an arginine from mischarged arginyl-tRNA$^{Lys}$ to a peptide substrate[21]. To gain further insight into the structural basis for arginyl-tRNA recognition by ATE1, we performed structural comparison analysis with other known aminoacyl-tRNA transferases, including FemX, L-PGS, and LFTR[34–36]. Superposition of the *sc*ATE1-GNAT-core with

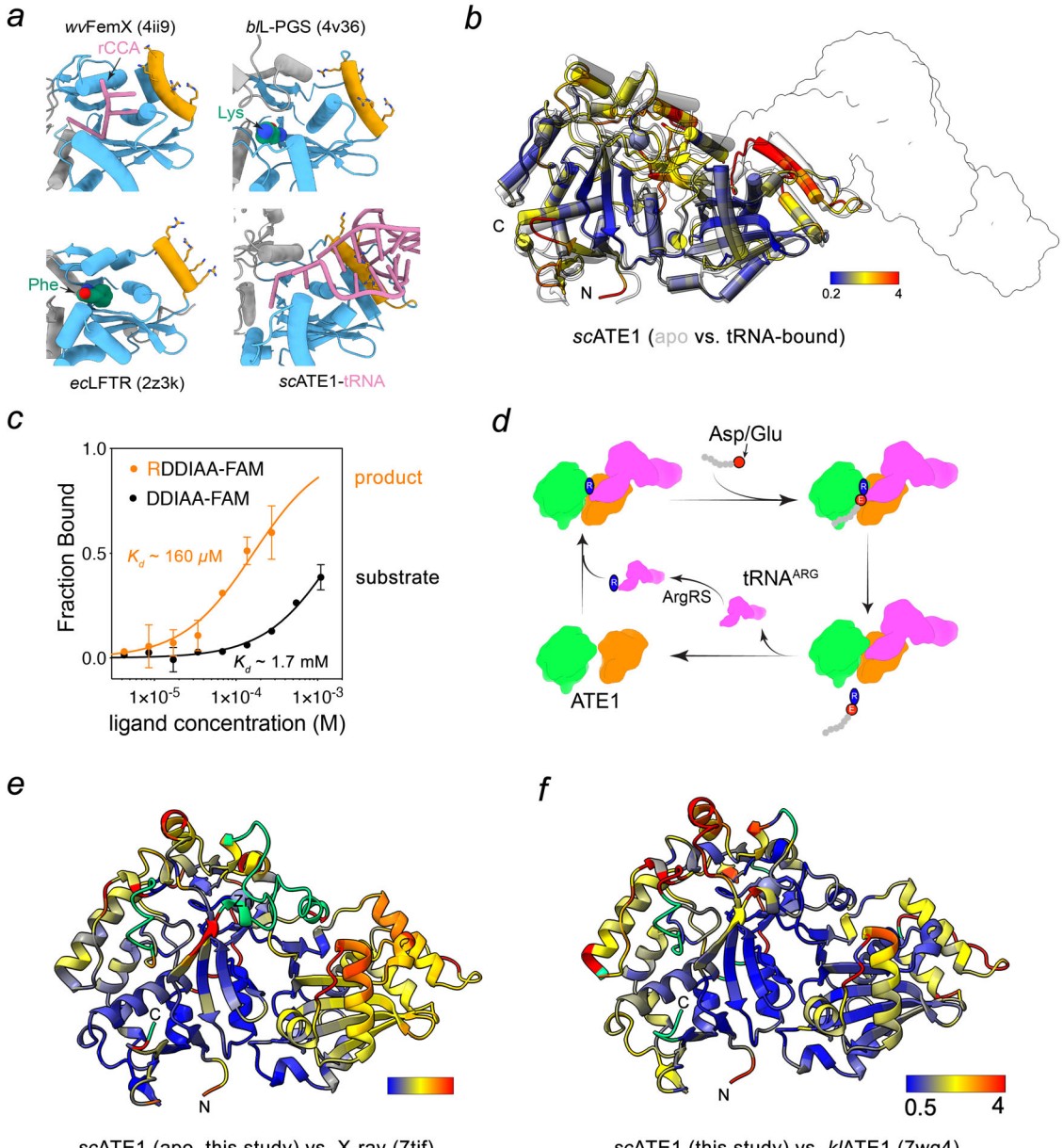

**Fig. 6 | Acceptor arm recognition and the proposed mechanism of ATE1-mediated arginylation. a** Ribbon diagrams of the structures of known aminoacyl-tRNA transferases including FemX (PDB: 4II9), ATE1, L-PGS (PDB: 4v36), and LFTR (PDB: 2z3k) in complex with indicated ligands shown in stick (violet). $\alpha f$ of ATE1 and corresponding tRNA interacting helices are colored in yellow, whereas the conserved GNAT-fold is shown in blue. **b** Overlay of the structures of ATE1 in apo (light grey) and tRNA-bound (color by RMSD calculated at each C$\alpha$) states. The surface of tRNA is shown in white. The color bar indicates the spectrum representing distances between aligned C$\alpha$ atom pairs, with blue and red specifying 0.2 and 4 Å, respectively. **c** Binding curves used to determine the $K_d$ values of ATE1-peptide interaction in a direct binding assay by MST. Error bar represents s.d. in triplicate measurements. **d** The proposed catalytic mechanism of protein N-terminal arginylation by ATE1. **e, f** The cryoEM structure of apo-ATE1 (this study) color by RMSD calculated at each C$\alpha$ compared with indicated X-ray structures of $sc$ATE1 (**e** PDB: 7tif) and $kl$ATE1 (**f** PDB: 7wg4). Green color indicating residues that are missing in the X-ray structures. The color bar indicates the spectrum representing distances between aligned C$\alpha$ atom pairs, with blue and red specifying 0.5 and 4 Å, respectively.

these transferases identifies a general mechanism of aminoacyl-tRNA recognition that diverges for the acceptor substrate sensing domains (Fig. 6a). Helix $\alpha f$ in ATE1 and its counterpart in related transferases includes a patch of positively charged Lys or Arg residues at the solvent-exposed surface to recognize the major groove of the tRNA acceptor arm. Additionally, our model indicates that the 3′ CCA of tRNA could be recognized by a hydrophobic patch on $sc$ATE1 (consisting of Phe, Tyr or Trp residues) in a manner that is similar to that observed in the $wv$FemX L-alanine transferase structure[34].

To determine the tRNA-induced conformational change on ATE1, we compared ATE1 structural models in apo-state and tRNA-bound

state with an arginine docked into the putative arginine binding pocket (denoted as ATE1•tRNA•Arg complex). Morphing between the apo $sc$ATE1 and an ATE1•tRNA•Arg complex model demonstrates a collective structural rearrangement of ATE1 (Fig. 6b and Supplementary Movie 1). Upon tRNA binding, helix $\alpha f$ of $sc$ATE1 slides into the major groove of the tRNA acceptor arm, while helices $\alpha h$ and $\alpha i$ shift closer to the substrate binding pocket. This motion likely promotes tighter binding of the substrate peptide to properly maintain its ideal positioning in the active site. In support of this notion, both the peptide substrate and the arginylated product bind to apo-ATE1 with much weaker affinities (Fig. 6c; $K_d$ ~ 1700 and 160 µM, respectively),

suggesting that arginyl-tRNA binding precedes substrate binding. Based on data from us and others, we thus propose a detailed catalytic cycle of ATE1-mediated protein arginylation reaction (Fig. 6d). Intracellular ATE1 displays a high affinity for arginyl-tRNA$^{Arg}$ and leads to the formation of ATE1•arginyl-tRNA$^{Arg}$. Binding to arginyl-tRNA$^{Arg}$ induces a conformational change in ATE1 and increases its affinity to acceptor substrates, which binds weakly to free ATE1. The arginine is then transferred from tRNA to the protein substrate, generating arginylated protein and uncharged tRNA. The reduced affinity of tRNA following the arginine transfer reaction facilitates its release, where subsequent recharging by ArgRS completes the catalytic cycle.

## Discussion

In this study, we have determined the recognition mechanism of ATE1 for the tRNA$^{Arg}$ cofactor to prime the complex for protein arginylation. Through its GNAT-fold subdomain, ATE1 primarily interacts with the major groove of the acceptor arm and the 3′-CCA end of tRNA. We show that a conformational change of helix $\alpha f$ of ATE1 occurs upon tRNA binding, which could be coupled with 3′-end arginine recognition as well as transmitting signals to the substrate binding domain. This mechanism is in stark contrast to the translational elongation factors, which bind the minor grooves of aminoacyl-tRNA acceptor arm and T-helix (Fig. 4) to deliver all tRNAs in an unbiased way. Furthermore, the tRNA interacting surface of ATE1 and elongation factors partially overlaps, indicating that protein arginylation and ribosomal translation are competitive processes. It is likely that the N-degron pathway could be activated when substoichiometric concentrations of elongation factors are insufficient to occupy all arginyl-tRNA$^{Arg}$, thus providing a delicate sensing mechanism to toggle between protein synthesis and degradation. Depiction of tRNA and ATE1 interactions not only sheds light on devising HIT-seq experiment[37] to decode potential sequence specificity within the acceptor arm but also lightens up the potential of using RNA oligos as therapeutics to inhibit ATE1.

How does ATE1 select arginyl-tRNA$^{Arg}$ from the large tRNA pool? We show that the A- and T-arms of tRNA are critical for mediating specific interactions with ATE1. Compared to non-substrate tRNAs such as tRNA$^{Lys}$, substrate tRNA$^{Arg}$ associates with ATE1 ~20-fold stronger (Fig. 5e). Our results help to explain why a mischarged arginyl-tRNA$^{Lys}$ did not support the arginylation reaction as reported by Avcilar-Kucukgoze et al.[21]. Because the tRNA$^{Arg}$ acceptor arm is sufficient for ATE1 binding, our results also support that tRNA-derived fragments containing the acceptor arm can mediate arginylation, as reported recently[21]. Additionally, we were able to dock a free arginine into the putative aminoacyl-binding pocket close to the last nucleotide A77 in our ATE1•tRNA structure (Supplementary Movie 1). We reasoned that the shape of this binding pocket would provide another layer of selectivity toward the arginyl group, which could explain why a mischarged lysyl-tRNA$^{Arg}$ did not support the arginylation reaction[21]. Future studies on ATE1/arginyl-tRNA$^{ARG}$ complex will be needed to fully understand the mechanisms of arginine transfer and selectivity toward arginyl-tRNA$^{Arg}$, particularly within a cellular context.

Another distinct feature of ATE1 is the presence of a strictly conserved 'CX$_2$CX$_n$CC' sequence motif, which has previously been reported to coordinate either an iron or a zinc ion[38,39]. We identified both zinc and calcium ions in purified scATE1 and showed that the cysteine cluster is an atypical zinc finger critical for the stability and function of ATE1. During the preparation of this manuscript, two crystal structures of apo ATE1 from budding yeast (*Saccharomyces cerevisiae* and *Kluyveromyces lactis*) were reported and are consistent with our cryo-EM structure as evidenced by an RMSD value of ~1.2 Å (Fig. 6e, f)[40,41]. In scATE1, several regions, including the ZnF, were not modeled into the crystal structure but could be resolved in our structures (Fig. 6e, colored in green) and were previously suggested to coordinate a [Fe-S] cluster[42]. While the cryoEM density is insufficient to accurately identify the metal within the cluster, we did not detect Fe in the EDTA-

extracted metal pool of our experiments (Fig. 2d). Furthermore, recombinant scATE1 was active after extensive purification procedures in ambient air that are detrimental to preserve [Fe-S] clusters, suggesting that zinc-coordination is sufficient for ATE1 activity. Consistent with our results, klATE1 was also shown to bind zinc, which was identified using inductively coupled plasma-mass spectrometry. Importantly, equivalent residues in the identified Arg-tRNA$^{Arg}$ binding pocket on scATE1 were critical for klATE1 activity in vitro, highlighting the conserved tRNA recognition mechanism proposed here (Fig. 6b). Future studies are needed to determine whether and how [Fe-S] cluster is incorporated into ATE1 and what functional significance is associated with its binding.

Collectively, our study provides the first glimpse into the specific substrate recognition mechanism of ATE1 and sheds light on strategies for modulating ATE1 activity. Moreover, our study builds a general framework for understanding other tRNA-dependent transferases. Lastly, since hijacking tRNAs from translation for regulatory functions is emerging as a new focus in normal physiology and disease[43], the unique tRNA interaction mode reported here provides a strategy to design specific interactors for aminoacyl-tRNA.

Protein arginylation was first identified over 60 years ago, yet its mechanism and function have remained largely elusive and controversial[3,16,44–46]. Our study has focused on the well-established ATE1-mediated N-terminal arginylation and protein degradation function, but many more questions exist. For instance, some ATE1 substrates, including actin and calreticulin, are not degraded following N-terminal arginylation[16,17,47,48]. Furthermore, ATE1 has been shown to mediate non-canonical midchain arginylation in which arginine is transferred to internal glutamate or aspartate of the protein sequence[18]. In these cases, arginylation was thought to stabilize target proteins or modulate their oligomeric states rather than promote protein turnover. Future studies in these areas will be needed for a better understanding of the mechanism of ATE1-mediated midchain arginylation and detailed functional roles of non-degradable arginylated proteins.

## Methods

### Protein expression and purification

The scATE1 was cloned into a pGEX 6p-1 vector with an N-terminal GST-tag and PreScission cleavage site and expressed in BL21(DE3)-RIL cells (Agilent). This RIL strain contains a rare codon plasmid encoding extra copies of *argU*, *ileY*, and *leuW* and overexpresses tRNA$^{Arg}$. Protein production was induced with 0.4 mM IPTG overnight at 16 °C in Luria broth (LB). The GST-tagged scATE1 proteins were purified on glutathione Sepharose 4B beads (GE Healthcare) in 50 mM Tris (pH 7.5), 0.5 M NaCl, 1 mM MgCl$_2$, and 5 mM DTT. The GST tag was cleaved with PreScission protease overnight at 4 °C. The pET21a ΔNHsArgRS plasmid with an N-terminal His-tag is a generous gift from Aaron Smith lab at University of Maryland. The His-tagged proteins were purified on Ni-NTA beads (Qiagen) in 50 mM Tris-HCl (pH 7.5) buffer, supplemented with 0.5 M NaCl, 5 mM β-ME, and 10 mM Imidazole. The His-tag was cleaved overnight at 4 °C with TEV protease. Proteins were further purified by size exclusion chromatography and concentrated in Millipore concentrators. All mutants were generated by site-directed mutagenesis using the Q5 polymerase mutagenesis protocol, grown, and purified as WT proteins.

### NMR experiments

NMR experiments were carried out at 298 K on Bruker 800 MHz spectrometers to identify the metal ions bound with scATE1. Buffer exchange was done with 10 mM ammonium acetate in D$_2$O for purified 500 μl of 30 μM ATE1 to remove small molecules including Tris and DTT. Unbound salts and metal ions were removed from the protein sample by ultrafiltration. Then the protein and flow through (FT) samples were freeze-dried. 200 μL of EDTA-protein and EDTA-FT

samples were then prepared by adding EDTA in $D_2O$ to a final concentration of 120 μM. The samples were heated to release ions and centrifuged. The supernatant consisting of EDTA-metal extraction was used to obtain the 1D proton NMR spectrum. The 200 μL metal-EDTA standard samples (pH 7) were prepared in $D_2O$ by adding metals $Fe^{2+}$, $Mg^{2+}$, $Ca^{2+}$, and $Zn^{2+}$ to the final concentration of 30 μM and EDTA to 120 μM. NMR data were collected using Topspin and analyzed using NMRDraw. The 1D proton NMR spectra obtained from each metal-EDTA standard were compared with the EDTA-metal extraction from the protein.

## Fluorescence spectroscopy

Spectra were recorded at 27 °C on a Molecular Devices Spectramax iD3 multimode microplate reader. The samples containing 1 μM WT scATE1 and increasing concentrations of the tRNA acceptor arms were excited at 270 nm. Experiments were performed in a buffer containing 20 mM Tris (pH 7.5), 125 mM NaCl and 3 mM DTT. Emission spectra were recorded over a range of wavelengths between 320 and 370 nm with a 2 nm step size and a 140 s integration time and averaged over 3 scans. The $K_D$ values were determined using a nonlinear least-squares analysis and the equation:

$$\Delta I = \Delta I_{max} \frac{\left( ([L]+[P]+K_d) - \sqrt{([L]+[P]+K_d)^2 - 4[P][L]} \right)}{2[P]} \quad (1)$$

where [L] is the concentration of the peptide, [P] is the concentration of ATE1, $\Delta I$ is the observed change of signal intensity, and $\Delta I_{max}$ is the difference in signal intensity of the free and bound states of ATE1. The $K_D$ value was averaged over three separate experiments, with error calculated as the standard deviation between the runs.

## Fluorescent microscale thermophoresis (MST) binding assay

The MST experiments were performed using a Monolith NT.115 instrument (Nanotemper) as described previously[49]. All experiments were performed with the purified scATE1 in a buffer containing 10 mM Tris, 5 mM sodium phosphate buffer (pH 7.0), 100 mM NaCl, 3 mM DTT, and 0.08% Tween-20. The final concentration of the fluorescein-labeled DNA(FAM-DNA) was 80 nM. Dissociation constants for the interaction of scATE1 with unlabeled tRNA acceptor arms were measured using a displacement assay in which increasing amount of unlabeled RNA were added into a preformed scATE1:FAM-DNA complex prepared by supplementing 30 μM ATE1 into each sample. The measurements were performed at 50% excitation power and medium MST power with 20 s MST on time and 1 s off time. For all measurements, samples were loaded into standard capillaries and 900–1300 counts were obtained for the fluorescence intensity. The $K_d$ and $IC_{50}$ values were determined with the MO. Affinity Analysis software (NanoTemper Technologies GmbH), using two independent MST measurements. The $K_i$ values for unlabeled tRNA micro helices with scATE1 were determined from the $IC_{50}$ values observed in the displacement assay and converted by the following equation:

$$K_i = [I]_{50} / \left( \frac{[L]_{50}}{K_d} + \frac{[P]_0}{K_d} + 1 \right) \quad (2)$$

where $[I]_{50}$ is the concentration free unlabeled ligand at 50% binding, $[L]_{50}$ is the concentration of free labeled DNA at 50% binding. The $K_d$ value is the dissociation constant of labeled DNA determined in the direct binding experiment described above. Measurements for all RNAs were done in triplicates.

## Cryo-electron microscopy data collection and image processing

Purified scATE1 protein at 0.3 mg/mL was applied onto a freshly glow discharged UltraAuFoil R1.2/1.3 grid (300 mesh) and plunge frozen using a vitrobot Mark IV (Thermo Fisher) with a blot force of 0 and 2 s blot time at 100% humidity at 4 °C. The cryoEM datasets were collected at National Cryo-Electron Microscopy Facility (NCEF). Movie stacks were recorded using an FEI Titan Krios transmission electron microscope G3i operated at 300 keV and equipped with a Gatan K3 direct electron detector and Gatan BioQuantum image filter operated in zero-loss mode with a slit width of 20 eV. Automated data collection was carried out using Latitude at a nominal magnification of 105,000x with a physical pixel size of 0.872 Å/pixel (0.436 Å/pixel at super-resolution) and a nominal dose of 40 e/Å$^2$ using a dose rate of 13.86 electron/pixel/second. 3.3-second exposures were fractionated into a total of 50 frames. The number of movies for a specified dataset was listed in Table S1. The defocus range was set to between −0.5 and −2.5 μm.

Image processing was performed using CryoSPARC version 3.3.2[50,51]. Both apo- and tRNA-bound structures were solved in the same dataset. The particles were automatically picked using the blob picker with an 80 Å diameter. Reference-free 2D classification was performed in streaming with 200 classes and limited maximum resolution to 18 Å. 2D class averages with expected size and shape were subjected for another round of 2D classification with 200 classes with maximum resolution set to 6 Å. After second round of 2D classification, particles from classes with resolution better than 10 Å and ECA less than 2 were selected for subsequent analysis. Additionally, particles were manually separated into two classes: one class with nucleic acid-like feature and one class without. These 2D classes were submitted for a "rebalance 2D" job type to trim particles from dominant views. The rebalanced particle set was then used for ab-initio reconstruction to generate the initial volume allowing using maximum resolution to 6 Å. 3D refinement was first performed using non-uniform 3D refinement with an initial lowpass resolution of 8 Å to preserve map features reconstituted in the ab initio 3D volume. Local refinement (NEW! job type) was subsequently performed with the following options turned on: (1) Use pose/shift gaussian prior during alignment, (2) Re-center rotation each iteration, (3) Re-center shifts each iteration, (4) Force re-do GS split and (5) FSC Noise-Substitution. The gold-standard Fourier shell correlation (FSC) of 0.143 criterion was used to report the resolution and the FSC curves were corrected for the effects of a soft mask using high-resolution noise substitution[52].

## Model building and refinement

AlphaFold II predicted scATE1 structure was used as the initial model[53]. Zinc atom was placed to coordinate cystine residues 20, 23, 94 and 95 in Coot (version 0.9.6)[54], and metal coordination restrains were generated using phenix.metal_coordinate[55]. The model was refined into the final cryoEM map using phenix.real_space_refine[55]. All statistics for structural models were reported in Table S1. Fig. panels depicting cryoEM maps or atomic models generated using Pymol and ChimeraX version 1.4[56]. Q-scores were calculated in MapQ from segger in Chimera version 1.6[57,58]. Maps colored by local resolution were generated using RELION 3.1[59].

## Molecular dynamic (MD) simulations

CryoEM structure of scATE1 with and without Zinc was used as the starting structure for MD simulations. Input files for MD simulations were prepared using tleap[60]. MD simulations were performed using the NAMD[61] and the amber ff19sb[62], ZAFF for Zinc[63], ions with the TIP3P water model[64]. Proteins were solvated in a cubic water box with a 16 Å padding in all directions. Sodium ions and chloride ions were added to achieve a physiological salt condition of 150 mM. The systems were energy minimized for 10000 steps to remove bad contacts. Then, the systems were equilibrated with all heavy atoms restrained harmonically and the temperature raised 10 K per 10000 steps starting from 0 to 300 K using temperature reassignment. After reaching the desired temperature, harmonic restraints were gradually reduced using a scale from 1.0 to 0 with a 0.2 decrement for every 50,000 steps. MD

simulations were performed under the NPT ensemble[65,66]. Langevin dynamics was used for constant temperature control, with the value of Langevin coupling coefficient and the Langevin temperature set to 5 ps and 300 K, respectively. The pressure was maintained at 1 atm using the Langevin piston method with a period of 100 fs and decay times of 50 fs. A time step of 2 fs was used for all the simulations by using the SHAKE algorithm[67] to constrain bonds involving hydrogen atoms. Analysis of trajectories were performed using scripts in VMD 1.9.6[68].

### Molecular dynamic flexible fitting (MDFF) of ATE1-tRNA$^{ARG}$ complex

Structure of tRNA$^{ARG}$ with the sequence 5′-GCGCCCUUAGCUCAGU UGGAUAGAGCAACGACCUUCUAAGUCGUGGGCCGCAGGUUCGAAUC CUGCAGGGCGCGCCA-3′ was modeled using PDB entry 1F7U as the starting template. Both the modeled tRNA and apo-form of scATE1 structure were docked into cryoEM map as rigid body, then the tRNA structure were manually shift away from protein to avoid steric clashes in the starting structure. Then all-atom MDFF simulations[69] with explicit solvent model were performed starting from this starting structure. The system was built with amber ff19sb[62] for protein, RNA.OL3 for RNA[70], ZAFF for zinc[62] and ions with the TIP3P water model[64]. The system was placed in a rectangle water box with 16 Å padding and charge-neutralizing sodium ions with additional salt (Na$^+$, Cl$^-$) around the solute resulting in 150 mM salt condition using tleap[60]. Input files for MDFF were prepared using the mdff package in VMD 1.9.6[68] with gscale=0.3 and sampled for 100 ns using NAMD3[61] on two NV-link NVidia A100 GPU cards. The variable SCALING_1_4 was set to 0.8333333 to be compatible with amber force field. Collective variable calculations (colvars) module in NAMD[71] was used to constrain the distance between the end of Lysine 304 and O3′ of A77 in tRNA to with a distance of 3 Å with a force constant of 10 kcal/mol/Å$^2$. Cross-correlation values for snapshots from the MD trajectory with cryoEM map were calculated using Situs package[72]. A frame from the last 10 ns of the trajectory with a cc value of 0.75 was selected and subsequently minimized using Geometry Minimization in phenix[55].

### Differential scanning fluorimetry

Spectra were recorded on a CFX96 Touch real-time PCR detection system (Bio-Rad). The reaction volume was 30 µL containing 5-10 µM scATE1 in 20 mM Tris (pH 7.5), 150 mM NaCl, 5 mM DTT, and SYPRO orange (Thermo Fisher 5,000X) to a final concentration of 2.5-5X with or without possible ligands in variable concentrations. The thermal gradient was conducted between 25 and 90 °C with 0.2 °C/min intervals.

### Gel shift assays (EMSA)

Increasing amounts of scATE1 were incubated with 1 µM tRNA acceptor arms, corresponding DNA minihelix, or 0.25 µM IVT tRNA$^{Lys}$ in 20 mM Tris-HCl (pH 7.5), 150 mM NaCl, 5 mM DTT for 30 min on ice. The reaction mixtures were loaded on 4–12% non-denaturing gradient gels (Invitrogen) and electrophoresis was performed in 1x TB buffer at 100 V for 3 h at 4 °C. Gels were stained with SYBR green and visualized with Biorad Gel Doc XR Imaging System. For binding with in vitro transcribed tRNA$^{Arg}$, scATE1-tRNA equilibrium binding reactions was performed at 30 °C (20 mM Tris 7.5, 150 mM NaCl, 0.001% NP-40, pH 7.5, 5′ end radiolabeled human tRNAs: 1 nM) with titrated scATE1 concentrations. RNA and protein were incubated for 10 min. Samples were then loaded on 8% non-denaturing PAGE (acryladmide:bis 19:1, 0.5X TBE). Gels were dried, and radioactivity in bound and free RNA was analyzed using Phosphorimager (GE) and ImageQuant 8.2.0 software (GE Healthcare, IL).

### Reporting summary

Further information on research design is available in the Nature Portfolio Reporting Summary linked to this article.

## Data availability

The cryo-EM maps have been deposited in the Electron Microscopy Data Bank under accession codes EMD-27871 and EMD-29638. The atomic coordinates for the deposited map have been deposited in the Protein Data Bank under accession codes 8E3S and 8FZR. Source data are provided with this paper. Other data are available from the corresponding author upon request. The structural data used in this study are available in the PDB database under accession codes 1F7U, 1B23, 1OB2, 2FMT, 4ii9, 4v36, 2z3k, 7tif, and 7wg4. Source data are provided with this paper.

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

## Acknowledgements

We thank Dr. Ebmeier at the University of Colorado Boulder Mass Spectrometry Facility for help with collecting and analyzing samples. We are grateful to Dr. Miyagi and Dr. Merrick for stimulating discussions and critical reading the manuscript. This work was supported by grants from NIH R00 CA241301 to Y.Z., R01 GM133841, RM1 GM142002, and R01 CA240993 to D.J.T and Case Comprehensive Cancer Center (P30CA043703 to Y.Z. and D.J.T.). We are grateful to the Cryo-Electron Microscopy Core at the CWRU School of Medicine and K. Li and K. Whiddon for access to the sample preparation and cryo-EM instrumentation. Computational support was provided by the Case Western Reserve University High-Performance Computing Cluster. This research was, in part, supported by the National Cancer Institute's National Cryo-EM Facility at the Frederick National Laboratory for Cancer Research under contract HSSN261200800001E. This publication was made possible by the Clinical and Translational Science Collaborative of Cleveland, UL1TR0002548 from the National Center for Advancing Translational Sciences (NCATS) component of the National Institutes of Health and NIH roadmap for Medical Research. Its contents are solely the responsibility of the authors and do not necessarily represent the official views of the NIH.

## Author contributions

T.A., W.H., X.Y., A.N., and X.L. performed experiments and together with E.J., D.T. and Y.Z. analyzed the data. W.H., D.T. and Y.Z. wrote the manuscript with input from all authors.

## Competing interests

The authors declare no competing interests.
