## [Peer Review File · Nature Communications]

The structural basis of tRNA recognition by arginyl-tRNA-protein transferaseREVIEWER COMMENTS

Reviewer #1 (Remarks to the Author):

Numerous non-canonical functions of tRNAs have been described, including in vitamin biosynthesis, heme biosynthesis and even membrane remodeling. Still, one of the most interesting ones, is the use Arg-tRNA^{Arg} in protein modification. Protein arginylation, targets proteins for degradation and is mediated by the enzyme ATE1 (arginyl-tRNA-protein transferase). Protein arginylation has been known for many years and yet the mechanism at play has remained elusive. The present manuscript by Abeywansa et al. Provide the structure of ATE1 as an apo-enzyme and as a substrate-bound enzyme, which revealed many interesting features of the enzyme including aspects of tRNA recognitions as well as an unusual Zn-coordinating fold. Overall, this is a very nice piece of work, and the experiments are carefully crafted and presented. With this said, there are several issues that need addressing. For example, the enzyme binds tRNA^{Arg} with high affinity (155 nM dissociation constant), how does this compare to non-substrate tRNAs in vitro? Also, what if the binding is performed with aminoacylated tRNA^{Arg}? Does it improve the binding further? Lastly, the authors conclude, using a stem-loop representative of acceptor arm of tRNA^{Arg}, that such a substrate is sufficient for binding. However, the dissociation constant is in the micromolar range. So, if anything, the real conclusion is that it can bind the stem-loop very poorly and in fact other domains of the tRNA may contribute to recognition. To claim that the stem loop is sufficient, they must perform enzyme assays and show that the mini-substrate, if bearing Arg, can be used for arginylation. In passing, it known that the acceptor stem coaxially stacks with the T-psi-C loop, did the authors test such a substrate? I expect binding will improve. Lastly, the authors talk about "catalytic mechanism" and yet no detail enzymatic analysis is provided. This is needed if the authors claim that they for the first time revealed the enzyme mechanism.

Minor comment:

In figure 1. Panel (a) I believe an arrow is missing towards donating Arg to "ribosomal translation".

Reviewer #2 (Remarks to the Author):

This is a very interesting and timely paper on ATE1 structures, following two recent publications that solved this structure using ATE1 from different yeast species. Here, the authors address the mechanisms of ATE1 catalysis and tRNA interactions, and propose general principles of ATE1 action in vivo.

ATE1 structure has eluded researchers for almost 30 years, and therefore, while this paper is not the first to solve this structure, it is novel and interesting enough to merit publication. However, I believe the authors need to do a better job putting their results in context of other studies in the field.

1. Please discuss the detailed correspondence of the structure solved here with the two recently published Ate1 structures. The similarities and differences, as well as any novel advances of the current study compared to the previous two, need to be prominently highlighted.
2. This paper discusses arginylation solely as an N-end rule degradation mechanism. Please add a discussion of non-degradatory ATE1 targets.
3. In addition to N-terminal arginylation, side chain arginylation has also been demonstrated. Can the authors speculate on how the side chain arginylation happens? If it is difficult to reconcile with Ate1 structure, can the authors predict the relative

frequency of side chain versus N-terminal arginylation? Can the authors comment on whether one or the other is likely to happen on protein versus peptide targets?

4. To me it was not clear from the models presented how ATE1 would distinguish between Lys- charged and Arg-charged tRNA. Given that ATE1 broadly recognizes tRNA from different species, does this mean it only interacts with the Arg-conjugated acceptor sequence? If so, how can ATE1 bind to tRNA so strongly as to copurify with it from cell extracts? This needs to be explained clearer.

5. Along the same lines, a recent study suggests that tRNA fragments can also work in the arginylation reaction. Can the authors discuss all these possibilities in more detail?

Reviewer #3 (Remarks to the Author):

This manuscript describes the cryoEM structures for ATE1 and for the ATE1-ArgtRNAArg complex. ATE1 transfers Arg residues to the N-ter of target proteins in a tRNA dependent manner. It belongs to the N-degron pathway that target proteins with N-ter degradation signals to proteolytic systems.

The task is challenging for the apo form of ATE1 with a relatively low molecular weight, but the authors report a 3.2 Å resolution map. For the ATE1-ArgtRNAArg complex the resolution estimate is around 6-8 Å.

The report could be of high interest, but unfortunately, a recent crystallographic structure for ATE1 from yeast *Kluyveromyces lactis* (ref. #36 in the manuscript) obscures the novelty for the structure of ATE1 in apo form. This leaves the structure for the ATE1-ArgtRNAArg complex the main potential contribution, but the current work falls short in providing good structural data for this complex. Actually, the story about the cryoEM sample of ATE1-ArgtRNAArg is difficult to follow. There are several points that weaken the quality of the work:

1.-In the main text it is said on lines 162-163:

“that we may obtain ATE1•tRNAArg complex suitable for structural interrogation using cryoEM by co-overexpressing tRNAArg with scATE1.”

This co-overexpressing is not documented in the Method section.

2.-The cryoEM data for the ATE1•tRNAArg complex is not presented at any time, and there is only one cryoEM data table for the ATE1 apo form.

3.-The cryoEM map for this same ATE1•tRNAArg complex is barely presented (fig 3e) and has not been deposited in the EMDB.

4.-There is a clear issue with preferred orientations for the ATE1•tRNAArg complex, something that the authors acknowledge. Their cryoEM map has an FSC threshold that suggests a resolution of 3.7 Å, but the authors estimate that the real resolution is around 6-8 Å. This is not a burden to deposit the map, and the authors should have done so.

5.-At this moderate resolution, the main contribution of the work is very limited since the ATE1-tRNA interaction is the current contribution of the manuscript.

Minor points

i.-How different is the modeled structure for ATE1 derived from the cryoEM data from the one obtained using Alphafold2? A comparison with the ATE1 structure from ref. 36 is also needed.

ii.- A cryoEM complex of ATE1 and the RNAArg-derived miniRNA hairpin (experiments shown in Fig. 4) could be a good way to improve the quality of the structural data.

Reviewer #4 (Remarks to the Author):

The manuscript submitted by Abeywansa et al. reports on the cryo-EM structure of ATE1. The structure of ATE1 has been reported previously by Kim et al (<https://doi.org/10.1073/pnas.2209597119>) and Van et al (<https://doi.org/10.1016/j.jmb.2022.167816>). The structure presented is an addition to those previously published. The claim that the molecular mechanism by which ATE1 interacts with and selects arginyl-tRNA is novel has also been reported by Kim et al. While the experimental study is sound, the interpretations of molecular dynamics conducted are the weak points. The differences in RMSD and Rg are not large enough to confirm destabilization as a result of Zn. This might be because the simulations are too short at 300ns. Perhaps longer simulations could should this pronounced localized destabilization. The authors might want to extend their simulations to see this effect. The Rg of four cysteines is not informative and can be seen as cherry-picking. In light of these observations, the reviewer believes this work is more suitable for a specialized journal and is not for nature comms.

Minor points: It's cysteines and not cystines as mentioned in several places in the manuscript. Also, extended Figure 3e is RMSF and not RMSD as indicated in the figure legend.

We thank the Editor and Reviewers for the insightful and constructive comments, which were helpful in revising the presentation of our findings and in strengthening the manuscript that describes the study. The individual comments/critiques from all reviewers have been considered and incorporated into the revised text and further described in the point-by-point response below.

Reviewer 1: ... revealed many interesting features of the enzyme including aspects of tRNA recognitions ... this is a very nice piece of work, and the experiments are carefully crafted and presented...

We thank the reviewer for the positive feedback.

Comment 1: ... the enzyme binds tRNA^{Arg} with high affinity (155 nM dissociation constant), how does this compare to non-substrate tRNAs in vitro?

Author's response: To address this question, we have now compared the binding affinities of tRNA^{Arg} with a non-substrate tRNA^{Lys}. The non-substrate tRNA^{Lys} binds ATE1 ~20-fold weaker than tRNA^{Arg}. We have included this data in the new Fig. 5d and Suppl. Fig. 4c and revised the text on page 10 (pasted below) to present the new results.

“... tRNA^{Lys} acceptor arm bound ATE1 only slightly weaker with a K_d value of 4 μ M (Fig. 5c-d) ...full length tRNA^{Lys} bound to ATE1 with an affinity similar to that of the acceptor arm alone (Fig. 5d and Supplementary Fig. S4c), further substantiating that the A+T arms of tRNA confers the selectivity of ATE1 toward tRNA^{Arg}...”

Related to reviewer 2 comment 4.

Reviewer 1, Comment 2: Also, what if the binding is performed with aminoacylated tRNA^{Arg}? Does it improve the binding further?

Author's response: Our current structural model (Fig. 3e) has predicted that the 3'-CCA end of tRNA is placed in a negatively charged pocket in a manner that is similar to that of Phe and Lys in LFTR and L-PGS respectively (Fig. 6a). Based on this comparison, we would predict that aminoacylated tRNA should further enhance the binding of the charged tRNA and ATE1. Ideally, we could conduct this experiment to support this hypothesis and we agree with the reviewer that the results would be informative to our investigation. However, such an experiment involves several challenges that have precluded us from including them in the present study. The primary limitation is due to the instability of aminoacylated tRNAs (they become hydrolyzed within minutes in solution) (Hentzen et al., 1972). As such, the interpretation would be complicated due to inherent heterogeneity for each purified batch that would consist of different proportions of charged and uncharged tRNA, as well as aminoacyl-bound tRNA. Because of this restriction, we have begun to explore the use of a synthetic non-hydrolysable Arg-tRNA^{Arg} for the affinity measurement. However, since no such mimetics currently exist, the task of designing, synthesizing, validating, and scale-up production is nontrivial and these efforts extend beyond the scope of the present manuscript.

Reviewer 1, Comment 3: ... To claim that the stem loop is sufficient, they must perform enzyme assays and show that the mini-substrate, if bearing Arg, can be used for arginylation. In passing, it known that the acceptor stem coaxially stacks with the T-psi-C loop, did the authors test such a substrate? I expect binding will improve.

Author's response: This reviewer brings up an important point, which we have attempted to clarify. The enzymatic assay using the stem loop from *E. coli* tRNA^{Arg} has already been reported (Avcilar-Kucukgoze et al., 2020); these prior findings are better incorporated in our revised manuscript. Also, as suggested, we include in the revised manuscript a comparison for the binding affinity of the acceptor stem with or without the T-psi-C arm. We discovered that the T-psi-C arm does indeed improve the binding affinity by ~ 4-fold as shown by gel shift and fluorescence assays. These data are included in new Fig. 5 panels d-f and discussed on page 10 of the revised manuscript (pasted below).

“... as seen in the scATE1•tRNA^{Arg} complex structure (Fig. 4a), the T-arm of tRNA^{Arg} is also proximal to ATE1, which likely provides additional interactions to stabilize binding. Indeed, synthesized RNA comprising both the A- and T-arms (denoted as A+T arms) of tRNA^{Arg} enhanced ATE1 binding affinity by 4-fold, which is comparable to the binding affinity of full length tRNA (Fig. 5d-f) ...”

Reviewer 1, Comment 4: ... the authors talk about “catalytic mechanism” and yet no detail enzymatic analysis is provided. This is needed if the authors claim that they for the first time revealed the enzyme mechanism.

Author's response: We thank the reviewer for pointing out the inaccurate description of our study and we have revised the manuscript to better clarify the key findings of our work. In revising our manuscript, we have attempted to avoid interpretation of the putative catalytic mechanism, although we do describe our findings in the context of the catalytic cycle. This interpretation is supported, primarily, by the biochemical experiments of the arginylation reaction included in Figures 5 and 6.

Reviewer 1, Minor comment: In figure 1. Panel (a) I believe an arrow is missing towards donating Arg to “ribosomal translation”. – added. Thank you.

Reviewer 2: This is a very interesting and timely paper on ATE1 structures, ... it is novel and interesting enough to merit publication. ...

We thank the reviewer for the positive feedback.

Comment 1: Please discuss the detailed correspondence of the structure solved here with the two recently published Ate1 structures. The similarities and differences, as well as any novel advances of the current study compared to the previous two, need to be prominently highlighted.

Author's response: It is true that, while this manuscript was in its final stages of preparation, two crystal structures of yeast ATE1 were published. Both structures describe the apo form of ATE1 protein and are consistent with our cryoEM structure (also in the apo form). Importantly, however, our work includes the ATE1 protein in complex with cofactor tRNA. This structure uniquely describes conformational changes of ATE1 that are induced by tRNA binding. The two conformations of ATE1 (apo and tRNA bound), allow us to provide an elaborate depiction of the catalytic cycle of arginylation reaction that was missing from the recently published x-ray structures/studies. Additionally, we show that the tRNA-ATE1 interaction is distinct from that of all known tRNA-protein complexes. This comparison has allowed us to describe interactions that are important for the selective interactions and function of ATE1-tRNA complexes and, similarly, could not be interpreted from the prior studies that included only the apo structure of ATE1. All of these advances are better

articulated in the revised manuscript. Specifically, a detailed comparison of our structure and the two crystal structures is now included as new data presented in Figure 6 and discussed on pages 13 of the revised manuscript (pasted below).

“...two crystal structures of apo ATE1 from budding yeast (*Saccharomyces cerevisiae* and *Kluyveromyces lactis*) were reported, and are consistent with our cryo-EM structure as evidenced by an RMSD value of ~ 1.2 Å (Fig. 6e-f)^{41,42}. In scATE1, several regions, including the ZnF, were not modeled into the crystal structure, but could be resolved in our structures (Fig. 6e, colored in green) and were previously suggested to coordinate an [Fe-S] cluster. ... equivalent residues in the identified Arg-tRNA^{Arg} binding pocket on scATE1 were critical for *k*ATE1 activity *in vitro*, highlighting the conserved tRNA recognition mechanism proposed here (Fig. 6b). Future studies are needed to determine whether and how [Fe-S] cluster is incorporated into ATE1 and what functional significance is associated with its binding. ...”

Additionally, we have transitioned some of these important findings from the supplementary material in the original manuscript to a new figure in the revised manuscript (Figure 4). New data included in Figure 5-6 were also added to include quantitative measurements supporting the tRNA selectivity and the proposed catalytic cycle, which could only be determined with data stemming from our study.

Reviewer 2, Comment 2: This paper discusses arginylation solely as an N-end rule degradation mechanism. Please add a discussion of non-degradatory ATE1 targets.

Author’s response: as suggested, we included a brief discussion of non-degradative ATE1 targets on Page 14 (pasted below).

“...Our study has focused on the well-established ATE1-mediated N-terminal arginylation and protein degradation function, but many more questions exist. For instance, some ATE1 substrates including actin are not degraded following N-terminal arginylation^{16,17,48}. ... In these cases, arginylation was thought to stabilize target proteins or modulate their oligomeric states rather than promote protein turnover. Future studies in these areas will be needed for better understanding of the mechanism of ATE1-mediated midchain arginylation and detailed functional roles of non-degradable arginylated proteins.”

Reviewer 2, Comment 3: In addition to N-terminal arginylation, side chain arginylation has also been demonstrated. Can the authors speculate on how the side chain arginylation happens? If it is difficult to reconcile with Ate1 structure, can the authors predict the relative frequency of side chain versus N-terminal arginylation? Can the authors comment on whether one or the other is likely to happen on protein versus peptide targets?

Author’s response: as suggested, we now include a brief discussion of side chain arginylation (page 14). Our structure clearly demonstrates a peptide binding pocket that can accommodate N-terminal peptides for catalysis. We were intentionally cautious in our discussion, however, since little is known about the abundance and competition/cooperation of terminal versus side chain reactions in cells (particularly in yeast) and because we did not want to overinterpret putative substrate (N-terminal or side chain) interactions in the context of our new structures. Ultimately, although we are actively pursuing studies to address questions regarding side chain arginylation, we felt that it would be prudent to limit speculations that are unsupported by experimental data in the current manuscript.

Reviewer 2, Comment 4: To me it was not clear from the models presented how ATE1 would

distinguish between Lys- charged and Arg-charged tRNA. Given that ATE1 broadly recognizes tRNA from different species, does this mean it only interacts with the Arg-conjugated acceptor sequence? If so, how can ATE1 bind to tRNA so strongly as to copurify with it from cell extracts? This needs to be explained clearer.

Author's response: we thank the reviewer for pointing out the ambiguity in our original manuscript. As suggested, we have now expanded the analysis of tRNA selectivity by measuring the binding affinities between ATE1 and different tRNA fragments. We show that non-substrate tRNA such as tRNA^{Lys} associates with ATE1 with a weaker affinity (~20-fold reduction), indicating tRNA nucleotide sequences and tertiary structure, particularly within the acceptor and T arms, contributes to tRNA-ATE1 interactions. Furthermore, based on our docking model of the ATE1•tRNA•Arg, we speculate that the ATE1 binding pocket also provides enhanced selection for the arginyl-group. The new data are included in new Fig. 5 and discussed on Page 12 (pasted below). To further address this question, the structure of ATE1/arginyl-tRNA^{Arg} complex is required. Despite many efforts on our part, this structure will require creative means of stabilizing a complex that is amenable to structural investigation. We believe we can achieve this objective with synthetic non-hydrolysable analogs, but such an effort extends beyond the scope of the present study.

“... How does ATE1 select arginyl-tRNA^{Arg} from the large tRNA pool? We show that the A- and T-arms of tRNA are critical for mediating specific interaction with ATE1. Compared to non-substrate tRNAs such as tRNA^{Lys}, substrate tRNA^{Arg} associates with ATE1 ~20-fold stronger (Fig. 5e). Our results help to explain why a mischarged arginyl-tRNA^{Lys} did not support the arginylation reaction as reported by Avcilar-Kucukgoze *et al*²². Because the tRNA^{Arg} acceptor arm is sufficient for ATE1 binding, our results also support that tRNA-derived fragments containing the acceptor arm can mediate arginylation as reported recently²². Additionally, we were able to dock a free arginine into the putative aminoacyl-binding pocket close to the last nucleotide A77 in our ATE1•tRNA structure (Supplementary Movie 1). We reasoned that the shape of this binding pocket will provide another layer of selectivity toward the arginyl-group, which could explain why a mischarged lysyl-tRNA^{Arg} did not support the arginylation reaction²². Future studies on ATE1/arginyl-tRNA^{ARG} complex will be needed to fully understand the mechanisms of arginine transfer and selectivity toward arginyl-tRNA^{Arg}, particularly within a cellular context ...”

Reviewer 2, Comment 5: Along the same lines, a recent study suggests that tRNA fragments can also work in the arginylation reaction. Can the authors discuss all these possibilities in more detail?

Author's response: In line with this reviewer's suggestion, we show that the acceptor arm of tRNA is sufficient to bind ATE1, indicating that the tRNA fragments containing the acceptor arm can mediate arginylation. This finding is in full agreement with, and further described in, other reports. As suggested, we expanded the discussion of the revised manuscript to include tRNA fragments, which can be found on page 12 (pasted below).

“...Because the tRNA^{Arg} acceptor arm is sufficient for ATE1 binding, our results also support that tRNA-derived fragments containing the acceptor arm can mediate arginylation as reported recently²²...”

Reviewer 3: ... The report could be of high interest, but unfortunately, a recent crystallographic structure for ATE1 from yeast *Kluyveromyces lactis* (ref. #36 in the manuscript) obscures the novelty ... the story about the cryoEM sample of ATE1-ArgRNAArg is difficult to follow.

We thank the reviewer for the positive and constructive feedback. We have improved the data analysis and revised the manuscript to better describe and highlight the ATE1/tRNA structure (first of its kind).

Comment 1: -In the main text it is said on lines 162-163: “that we may obtain ATE1•tRNAArg complex suitable for structural interrogation using cryoEM by co-overexpressing tRNAArg with scATE1.” This co-overexpressing is not documented in the Method section.

Author’s response: We apologize for the oversight in omitting these important details in our original manuscript. We have edited the revised manuscript to include more details about the co-overexpression in the main text on page 8 (pasted below) and Method section.

“...we used *E. coli* BL21 (DE3)-RIL cells for protein production, which encodes a rare codon plasmid containing an extra copy of *argU* and overexpresses tRNA^{Arg}....”

Reviewer 3, Comment 2: -The cryoEM data for the ATE1•tRNAArg complex is not presented at any time, and there is only one cryoEM data table for the ATE1 apo form. **Comment 3:** -The cryoEM map for this same ATE1•tRNAArg complex is barely presented (fig 3e) and has not been deposited in the EMDB. **Comment 4:** -...This is not a burden to deposit the map, and the authors should have done so.

Author’s response: as suggested, we have updated the table, deposited the map in the EMDB, and included more details in describing the structure of the complex.

Reviewer 3, Comment 5: -At this moderate resolution, the main contribution of the work is very limited since the ATE1-tRNA interaction is the current contribution of the manuscript.

Author’s response: Since our original submission, we have further refined and improved the map resolution. Even though the resolution for the complex is not atomic, the tRNA binding mode can be confidently determined. Specifically, the resolution of the map is sufficient to identify the the major groove of the acceptor arm to accurately model the full tRNA molecule. Thus, we are able to interpret the interactions of ATE1 and the 3’-region of the tRNA acceptor arm (Figure 1) and to discuss how it differs from predicted models (Hebecker et al., 2015; Kim et al., 2022). Additionally, the tRNA-bound ATE1 complex differs in conformation when compared to the apo ATE1 structure. Our experimental data allows us to discuss these functional conformations and, again, describe differences that cannot be inferred from previous predictions (new Fig. 6a and Supplementary movie 1). Additionally, we are able to discuss the binding mode of tRNA with ATE1, which is different from all known tRNA binding partners including models of tRNA-dependent aminoacyl-transferases (new Fig. 4). Finally, our combined biochemical new structural data allow us to more accurately propose the catalytic cycle of arginylation reaction (Fig. 6d). This critical information will

guide future attempts to assemble substrate-bound ternary complexes and capture reaction intermediates.

In the revised manuscript, we include additional data (Fig. 4, 5 and 6) and have expanded the discussion on Pages 10 and 12 to better highlight the advances that are novel to our investigation. Also, please see response to Reviewer 2, Comment 1.

Minor points:

i.-How different is the modeled structure for ATE1 derived from the cryoEM data from the one obtained using Alphafold2? A comparison with the ATE1 structure from ref. 36 is also needed. – added to new Fig. 6e-f and discussion on Page 13.

ii.- A cryoEM complex of ATE1 and the RNAArg-derived miniRNA hairpin (experiments shown in Fig. 4) could be a good way to improve the quality of the structural data. – We eagerly attempted this suggestion in the hopes that it would, indeed, provide a means to improve the quality of our complex map. Unfortunately, however, we were unable to do so, primarily due to low occupancy (or dissociation during grid prep) for the hairpin in the sample. This structure will require creative means of sample stabilization and grid preparation, and remains to fall beyond the scope of the present study.

Reviewer #4, Comment 1: ... The structure of ATE1 has been reported previously by Kim et al and Van et al. The structure presented is an addition to those previously published. The claim that the molecular mechanism by which ATE1 interacts with and selects arginyl-tRNA is novel has also been reported by Kim et al.

Author's response: It is true that, while this manuscript was in its final stages of preparation, two crystal structures of yeast ATE1 were published. Both structures describe the apo form of ATE1 protein and are consistent with our cryoEM structure (also in the apo form). Importantly, however, our work includes the ATE1 protein in complex with cofactor tRNA. This structure uniquely describes conformational changes of ATE1 that are induced by tRNA binding. The two conformations of ATE1 (apo and tRNA bound), allow us to provide an elaborate depiction of the catalytic cycle of arginylation reaction that was missing from the recently published x-ray structures/studies. Additionally, we show that the tRNA-ATE1 interaction is distinct from that of all known tRNA-protein complexes. This comparison has allowed us to describe interactions that are important for the selective interactions and function of ATE1-tRNA complexes and, similarly, could not be interpreted from the prior studies that included only the apo structure of ATE1. All of these advances are better articulated in the revised manuscript. Specifically, a detailed comparison of our structure and the two crystal structures is now included as new data presented in Figure 6 and discussed on pages 12-13 of the revised manuscript. Additionally, we have transitioned some of these important findings from the Supplementary material in the original manuscript to a new figure in the revised manuscript (Figure 4). New data included in Figure 5 were also added to include quantitative measurements supporting the proposed catalytic cycle, which could only be determined with data stemming from our study.

Reviewer #4, Comment 2: ... the experimental study is sound, the interpretations of molecular dynamics conducted are the weak points. The differences in RMSD and Rg are not large enough to confirm destabilization as a result of Zn. This might be because the simulations are too short at 300ns. Perhaps longer simulations could should this pronounced localized destabilization. The authors might want to extend their simulations to see this effect. The Rg of

four cysteines is not informative and can be seen as cherry-picking.

Author's response: The goal of MD simulation here is to investigate the potential outcome of protein integrity in the absence of Zn binding. To avoid confusion, we have revised the text to emphasize that MD simulations are not intended to confirm the destabilization in the absence of Zinc, but rather to provide insights into the potential outcome when Zinc is not included. The destabilization in 'no zinc condition' is evident within the first 70 ns of the simulation and the immediate perturbation on ATE1 largely involves the putative substrate binding site. Additionally, this data was corroborated by the loss of tRNA binding in the absence of Zinc as revealed by EMSA experiments. The lack of significant changes in the global structural parameters, such as RMSD and Rg, demonstrates the reliability of the force field parameters. The Rg of four cysteines are not intended for 'cherry-picking' data but to highlight the local destabilization of the site surrounding the Zinc as compared to global changes throughout the protein. We therefore think longer simulation will not strengthen our claims.

Minor points: It's cysteines and not cystines as mentioned in several places in the manuscript. Also, extended Figure 3e is RMSF and not RMSD as indicated in the figure legend. – corrected, thanks.

References

- Avcilar-Kucukgoze, I., Gamper, H., Polte, C., Ignatova, Z., Kraetzner, R., Shtutman, M., Hou, Y. M., Dong, D. W., & Kashina, A. (2020). tRNA(Arg)-Derived Fragments Can Serve as Arginine Donors for Protein Arginylation. *Cell Chem Biol*, 27(7), 839-849 e4.
- Hebecker, S., Krausze, J., Hasenkampf, T., Schneider, J., Groenewold, M., Reichelt, J., Jahn, D., Heinz, D. W., & Moser, J. (2015). Structures of two bacterial resistance factors mediating tRNA-dependent aminoacylation of phosphatidylglycerol with lysine or alanine. *Proceedings of the National Academy of Sciences of the United States of America*, 112(34), 10691–10696.
- Hentzen, D., Mandel, P., & Garel, J. P. (1972). Relation between aminoacyl-tRNA stability and the fixed amino acid. *Biochimica et Biophysica Acta (BBA) - Nucleic Acids and Protein Synthesis*, 281(2), 228–232. [https://doi.org/10.1016/0005-2787\(72\)90174-8](https://doi.org/10.1016/0005-2787(72)90174-8)
- Kim, B. H., Kim, M. K., Oh, S. J., Nguyen, K. T., Kim, J. H., Varshavsky, A., Hwang, C.-S., & Song, H. K. (2022). Crystal structure of the Ate1 arginyl-tRNA-protein transferase and arginylation of N-degron substrates. *Proceedings of the National Academy of Sciences*, 119(31), e2209597119.

REVIEWERS' COMMENTS

Reviewer #1 (Remarks to the Author):

I believe the authors have done an excellent job of addressing all my concerns raised in the previous cycle. I commend them for clearly explaining all the questions raised and adding new experiments as needed.

Reviewer #2 (Remarks to the Author):

The authors addressed all my comments to my satisfaction. One small thing: in the section where they added non-degradatory functions of arginylation, in addition to the actin work, the work on calreticulin (from M. Hallak's lab) should be cited. It would also be good to mention the non-degradatory arginylation screens and a wealth of putative targets that have been reported over the years, possibly by citing a more recent review, not to make an impression that all arginylation is about degradation.

Reviewer #3 (Remarks to the Author):

The authors have addressed the raised points in the review. My main concern was related to the quality of the cryoEM map for the ATE1•tRNAArg complex. This density map has now been improved significantly as clearly seen in Fig. 3e (could the authors describe the image processing that improved this map?), and the density map and the derived atomic model have been deposited in the corresponding databases.

There is one question, though, that needs clarification. In the current version of the manuscript it seems that both cryoEM maps for ATE1 and ATE1•tRNAArg complex were calculated using the same dataset and sample and after separation of particles (averages?) after 2D classification, as suggested in:

On-line methods lines 79-80

"Additionally, particles were manually separated into two classes: one class with nucleic acid-like feature and one class without."

Is it correct? Or this sentence applies only to the dataset for the ATE1•tRNAArg complex?

If these are two different samples and datasets, these must be clarified in the Extended Data Table 1 with separated number of micrographs for each data set.

In this same issue, in:

On-line methods lines 3-5

"The scATE1 was cloned into a pGEX 6p-1 vector with an N-terminal GST-tag and PreScission cleavage site and expressed in BL21(DE3)-RIL cells (Agilent). This RIL strain contains a rare codon plasmid encoding extra copies of argU, ileY, and leuW and overexpresses tRNAArg."

It suggests that the expression of scATE1 was always intended to produce the ATE1•tRNAArg complex, and hence, that there was just one sample that produced the two maps after classification of the cryoEM data that separates tRNA bound from tRNA free ATE1. Is it right?

Thus, it is not clear whether the ATE1 and ATE1•tRNAArg are coming from independent or the same sample.

Reviewer #4 (Remarks to the Author):

The authors have responded satisfactorily to the questions raised by this reviewer.

We thank the Reviewers for the positive feedback. The additional comments/critiques have been considered and incorporated into the revised text and addressed in the point-by-point response below.

Reviewer #1 (Remarks to the Author): I believe the authors have done an excellent job of addressing all my concerns raised in the previous cycle. I commend them for clearly explaining all the questions raised and adding new experiments as needed.

Reviewer #2 (Remarks to the Author): The authors addressed all my comments to my satisfaction. One small thing: in the section where they added non-degradatory functions of arginylation, in addition to the actin work, the work on calreticulin (from M. Hallak's lab) should be cited. It would also be good to mention the non-degradatory arginylation screens and a wealth of putative targets that have been reported over the years, possibly by citing a more recent review, not to make an impression that all arginylation is about degradation. - added

Reviewer #3 (Remarks to the Author): The authors have addressed the raised points in the review. My main concern was related to the quality of the cryoEM map for the ATE1•tRNAArg complex. This density map has now been improved significantly as clearly seen in Fig. 3e (could the authors describe the image processing that improved this map?), and the density map and the derived atomic model have been deposited in the corresponding databases.

There is one question, though, that needs clarification. In the current version of the manuscript it seems that both cryoEM maps for ATE1 and ATE1•tRNAArg complex were calculated using the same dataset and sample and after separation of particles (averages?) after 2D classification, as suggested in:

On-line methods lines 79-80

“Additionally, particles were manually separated into two classes: one class with nucleic acid-like feature and one class without.”

Is it correct? Or this sentence applies only to the dataset for the ATE1•tRNAArg complex?

If these are two different samples and datasets, these must be clarified in the Extended Data Table 1 with separated number of micrographs for each data set.

In this same issue, in:

On-line methods lines 3-5

“The scATE1 was cloned into a pGEX 6p-1 vector with an N-terminal GST-tag and PreScission cleavage site and expressed in BL21(DE3)-RIL cells (Agilent). This RIL strain contains a rare codon plasmid encoding extra copies of argU, ileY, and leuW and overexpresses tRNAArg.”

It suggests that the expression of scATE1 was always intended to produce the ATE1•tRNAArg complex, and hence, that there was just one sample that produced the two maps after classification of the cryoEM data that separates tRNA bound from tRNA free ATE1. Is it right?

Thus, it is not clear whether the ATE1 and ATE1•tRNAArg are coming from independent or the same sample.

Author's response: Both apo- and tRNA-bound ATE1 structures were solved in the same sample. We have clarified it in the methods. Additional details were also provided regarding the image processing steps.

Reviewer #4 (Remarks to the Author): The authors have responded satisfactorily to the questions raised by this reviewer.